# CorticalFlow: A Diffeomorphic Mesh Deformation Module for Cortical Surface Reconstruction

**Léo Lebrat**[†]
CSIRO, QUT
leb026@csiro.au

**Rodrigo Santa Cruz**[†]
CSIRO, QUT
fon022@csiro.au

**Frédéric de Gournay**
IMT - UMR5219
degourna@insa-toulouse.fr

**Darren Fu**
UQ

**Pierrick Bourgeat**
CSIRO

**Jurgen Fripp**
CSIRO

**Clinton Fookes**
QUT

**Olivier Salvado**
CSIRO, Data61.

## Abstract

In this paper we introduce CorticalFlow, a new geometric deep-learning model that, given a 3-dimensional image, learns to deform a reference template towards a targeted object. To conserve the template mesh's topological properties, we train our model over a set of diffeomorphic transformations. This new implementation of a flow Ordinary Differential Equation (ODE) framework benefits from a small GPU memory footprint, allowing the generation of surfaces with several hundred thousand vertices. To reduce topological errors introduced by its discrete resolution, we derive numeric conditions which improve the manifoldness of the predicted triangle mesh. To exhibit the utility of CorticalFlow, we demonstrate its performance for the challenging task of brain cortical surface reconstruction. In contrast to current state-of-the-art, CorticalFlow produces superior surfaces while reducing the computation time from nine and a half minutes to one second. More significantly, CorticalFlow enforces the generation of anatomically plausible surfaces; the absence of which has been a major impediment restricting the clinical relevance of such surface reconstruction methods.

## 1 Introduction

The field of 3D shape reconstruction using deep learning techniques has attracted much attention. Recently, a plethora of methods have been developed for problems such as single-view object reconstruction [22, 50, 77], surface generation [27, 74], and meshing noisy point clouds [32, 42]. At first, these methods solely aimed to retrieve surface meshes as geometrically close as possible to the target shape. However, recent applications require generating regular meshes with a known genus, such as physics simulation, 3D-printing, and clinical analysis of anatomical surfaces [24, 58, 62].

In this direction, three approaches in the literature stand out: DeepCSR [61], Voxel2Mesh [73], and Neural Mesh Flow (NMF) [30]. DeepCSR first predicts implicit surface functions and then employs an iso-surface extraction method along with a topology correction algorithm to obtain genus-zero surfaces without handles or holes. Voxel2Mesh extends the vertex-wise template deformation approach of Wang et al. [71] by optimizing several mesh-smoothing penalty functions. In contrast, NMF builds an invertible mapping that enforces topology conservation upon the resolution of an Ordinary Differential Equation (ODE) through a sequence of residual blocks called Neural Ordinary Differential Equation (NODE) [9]. However, these methods come with several limitations. The topology correction algorithm employed by DeepCSR is computationally expensive and is blind

---

† Equal contribution
Check our project web-page https://lebrat.github.io/CorticalFlow/

towards the anatomical validity of its reconstructions, which can result in implausible corrections and mesh artifacts. On the other hand, Voxel2Mesh and NMF rely on time-demanding and vertex-dependent building blocks such as graph convolution that do not scale up well as the number of vertices in the template mesh increases to accommodate complex shapes. In addition, the dynamics learned by the NODE model in NMF can be very complex and may lead to a non-diffeomorphic mapping resulting in self-intersections in the reconstructed mesh.

This paper introduces CorticalFlow (CF), a new geometric deep learning model that smoothly deforms a template mesh towards complex shapes producing high-resolution regular meshes. First, a simple 3D convolution neural network predicts a dense 3D flow field from a volumetric image with a modest GPU memory footprint. Second, we formulate a tractable mathematical framework to compute diffeomorphic mapping for each vertex by solving a flow ODE. We derive sufficient and comprehensible conditions for meeting the diffeomorphic properties of these transformations. Finally, a sequence of these diffeomorphic mappings is composed to produce accurate high-resolution genus-zero regular meshes.

To evaluate our approach and compare it to existing techniques for regular surface reconstruction, we consider the problem of brain cortical surface reconstruction, which is an essential step for the analysis of brain morphometry in neurodegenerative diseases [18] and psychological disorders [57]. Given a 3D MRI of the brain, the goal is to describe the inner and outer surfaces of the brain cortex, which are both homeomorphic to a sphere. Cortical surface reconstruction is challenging given the complexity, high resolution, and regularity required for the predicted meshes. In our experiments, CorticalFlow is more accurate than state-of-the-art methods, providing an average reduction of $17.38\%$ in Chamfer distance across all cortical surfaces compared to DeepCSR (the second-best performing method in this criteria). In terms of surface regularity, it surpasses NMF or Voxel2Mesh with an average reduction of at least $32.58\%$ of self-intersecting faces while handling template meshes with many more vertices. It is also faster and more memory-efficient than all of these competitors.

## 2   Related Works

### 2.1   Geometric deep learning for surface reconstruction

Supervised surface reconstruction can be broadly categorized according to the 3D shape representation used to encode the prediction as either volumetric, implicit surfaces representation, novel geometric primitives, or geometric [22].

Volumetric methods predict shapes encoded as a 3D grid of voxels containing discretized surface representations such as occupancy [11] and level-sets [46]. From this representation, surfaces are obtained using iso-surface extraction methods, such as marching cubes [41]. While 3D volumetric processing is amenable to a convolutional neural network, the memory requirements are often a limitation to attain high-resolution reconstructions (it grows cubically with the voxel-grid resolution). To overcome this issue, approaches based on octrees [35, 68, 72] have been proposed to increase the output resolution from a voxel-grid of $32^3$ to $256^3$. Unfortunately, these approaches sacrifice speed and necessitate the redefinition of standard network operations such as convolution, pooling, and unpooling for this hierarchical data structure. Furthermore, as presented in [61], even at this level of resolution, the precision is too coarse to capture the highly curved regions of the cortical surfaces.

Implicit surface methods alleviate resolution limitations of the volumetric methods by directly predicting surface representations like occupancy [45], signed distance [54, 75], and 3D Gaussians [26] for points with continuous coordinates. This formulation allows synthesizing grids at an arbitrary resolution during inference with an easily implementable local refinement procedure, while training is performed stochastically over a small subset of sampled points. Following this approach, Santa Cruz et al. [61] proposed DeepCSR, the first geometric deep learning model for cortical surface reconstruction. Its main limitation is the difficulty to control the topology and mesh quality of the reconstructed surfaces, which hampers atrophy estimation used for neurodegenerative disease diagnosis [24, 58, 62]. As a result, DeepCSR resorts to a computationally expensive topology correction algorithm to produce a final cortical surface almost free of artifacts and with a spherical topology.

Methods based on geometric primitives build a surface representation to approximate complex shapes as the union of these primitives. In this category, we can highlight the works of Niu et al. [51] and Groueix et al. [28] which propose to approximate complex object shapes with a collection of

"cuboids" or "surface patches". Recent works by [10, 17] revisit the convex decomposition idea and propose to reconstruct complex object shapes by predicting collections of convex parts. The former predicts a set of localized convex polytopes formed by their hyperplane parameters and a translation vector, while the latter predicts a binary space partitioning tree to reconstruct the target shape. While very promising in terms of information compactness, these approaches are challenged to generate cortical surfaces due to their varied curvatures, which require a large number of convex parts to produce accurate results.

Finally, geometric methods comprise techniques that allow estimating a high-resolution mesh by transforming a known template mesh [52, 53, 65, 71]. Following this approach, Wang et al. [71] propose a graph-convolution network to predict vertex-wise deformations of a spherical mesh while dynamically increasing its resolution with a point pooling process. Wickramasinghe et al. [73] extended this model for the reconstruction of smooth anatomical surfaces such as the liver or hippocampus for different image modalities. Topological errors are reduced using three different penalty functions in the loss function. Recently, Gupta and Chandraker [30] leverage NODE blocks [9] to parameterize regular deformations that allow conserving the two-manifoldness property of the input template.

Indeed, the "*manifoldness*" measures as non-manifold edges or non-manifold vertices, and defined by Gupta and Chandraker [30], are conserved by a deformable model which is not generating new vertices since those properties are inherited from the template mesh (only the vertices' positions are affected). However, the normal consistency (non-manifold faces) is only conserved by homeomorphisms which can at most flip globally the faces' normal orientation.

## 2.2 Generation of diffeomorphic mappings

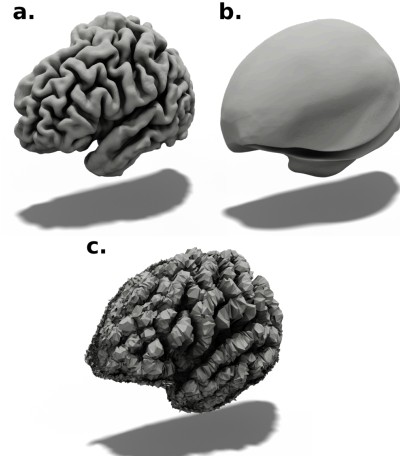

Figure 1: Generating regular meshes of the left hemisphere of the brain cortical surface. **a.** *CorticalFlow*, **b.** *NMF* with a global feature descriptor [30], **c.** *UNet* without diffeomorphic parametrization or geometric penalization.

The reconstruction of regular surfaces from a deformable model is a subtle trade off between finding the right parameterization or a suitable level of regularization during training. The delicacy of this problem is illustrated in Figure 1. First, one can employ multiple penalty functions as Chamfer normal, normal consistency, Laplacian loss, or edge length loss [71, 73]. It is worth mentioning that without such penalizations, a deformable model will learn non-smooth deformations, which leads to irregular meshes (as shown in Figure 1.c). However, those penalizations simply encourage the reconstructed surface to be regular. The second approach consists of parameterizing the set of learned deformations; this approach is favored in our paper since it allows stronger theoretical guarantees and is not subject to hyper-parameter tuning. A natural framework to generate invertible deformations is to consider a flow ODE [20, 69]. This framework has been successfully applied in pattern recognition and image registration [2, 3, 7, 19]. The main idea is to consider the mapping as the solution at time $\tau$ of an initial value problem (IVP) of the form,

$$\frac{\mathrm{d}\Phi(s;\mathbf{x})}{\mathrm{d}s} = v\left(\Phi(s;\mathbf{x}), s\right), \text{ with } \Phi(0;\mathbf{x}) = \mathbf{x}, \tag{1}$$

under a regularity hypothesis on $v$ and upon boundedness of its support, and using the Picard-Lidelöf theorem, one can show that a unique solution of this problem exists for $\tau \in \mathbb{R}$. In addition, the mapping $\mathbf{x} \mapsto \Phi(s;\mathbf{x})$ defines a family of diffeomorphisms [1, 6] for all time $s \in [0, \tau]$ whose inverse can be computed through a backward integration.

When the vector field $v$ is constant over time *i.e.* $v : \mathbb{R}^3 \to \mathbb{R}^3$, Equation (1) describes the Stationary Velocity Field (SVF) framework [1, 3]. If $v$ is a time-varying vector field, the framework described in (1) becomes the LDDMM model [5, 8, 64, 70].

This generic framework has been successfully applied within deep learning methods for diffeomorphic image registration [15, 47], point-cloud completion [49], single view reconstruction [30] and to parameterize set of deformations [37].

The SVF formulation has been particularly fecund in medical deep-learning registration [15, 40, 47, 76], where the resolution of Equation (1), is performed using the scaling and squaring method [2, 3] to predict the displacement of each voxel-center and to compute the registered image. However, this technique is not suitable for points that lie on non-regular coordinates. Naively, one can compute these mappings on a dense grid and then interpolate the deformation at non-regular coordinates. This simple approach is subject to two main limitations. Firstly, Equation (1) has to be solved in our context for millions of grid points where only a few hundred thousand vertices are displaced. Secondly, one cannot guarantee the invertibility of such a mapping with linear interpolation and one cannot compose provably several of those approximated mappings.

In [30, 49], the problem is solved using a black-box neural ODE [9] and by learning a neural vector field $v$. Despite allowing learning a time-dependent vector field, this approach has shown its limitations in our targeted application. Cortical surfaces are unique to each individual; indeed, the cortical folding patterns are similar to a fingerprint [44] and constitute a distinctive biometric for each individual. Moreover, we observe that the classical approach, which consists of conditioning the neural ODE on a global feature descriptor of the input image, fails to provide satisfactory results for cortical surface reconstruction (see Figure 1.b. and the supplementary material). Instead, one has to equip each moving vertex with a local feature descriptor of the input image, limiting the number of vertices of the resulting mesh.

Our work lies at the intersection of these methods. We propose to extend the SVF framework for points lying in real-coordinates, with particular care given to the numerical affordability of the ODE solver. We define a multi-scale approach, so that the final deformation is the result of the composition of three successive deformations that allow to approach more complex mappings and alleviate the limitations of the one vector field SVF framework [23, 43]. This framework is memory efficient, theoretically tractable, and can seamlessly handle large template meshes ($\approx 450$k vertices).

## 3 Method

CorticalFlow (CF) is a multi-level deep learning architecture composed of several Diffeomorphic Mesh Deformation (DMD) modules. It takes as input a 3-dimensional Magnetic Resonance Image (MRI) of a patient brain denoted $\mathbf{I}$ (tensor of dimensions $H \times W \times D$) and a template $\mathcal{T}_i$ (where $i$ represents the degree of refinement of the template). CorticalFlow outputs the surface representation of an anatomical substructure by composing stackable diffeomorphic deformations generated by DMD modules. CorticalFlow with $k$ deformations ($\mathrm{CF}^k$) can be written using the following recurrence,

$$\mathrm{CF}^1_{\theta_1}(\mathbf{I}, \mathcal{T}_1) = \mathrm{DMD}(\mathrm{UNet}^1_{\theta_1}(\mathbf{I}), \mathcal{T}_1))$$
$$\mathrm{CF}^{i+1}_{\theta_{i+1}}(\mathbf{I}, \mathcal{T}_{i+1}) = \mathrm{DMD}(\mathrm{UNet}^{i+1}_{\theta_{i+1}}(\mathbf{U}_1^\frown \cdots \mathbf{U}_i^\frown \mathbf{I}), \mathrm{CF}_i(\mathbf{I}, \mathcal{T}_{i+1})) \quad \text{for } i \geq 1, \qquad (2)$$

with $\mathbf{A}^\frown \mathbf{B}$ the channel-wise concatenation of the tensors $\mathbf{A}$ and $\mathbf{B}$ and where $\mathbf{U}_k$ denotes the output of the $k$-th $\mathrm{UNet}^k_{\theta_k}$ parameterized by $\theta_k$.

In our paper we describe $\mathrm{CF}^3$, a version of CorticalFlow with three stages where each stage is learned successively. CorticalFlow is trained in a supervised fashion, given a dataset $\mathcal{D}$ composed of pairs of MR-image $\mathbf{I}$ and triangle mesh $S$ representing a cortical structure and for $i \in \{1, 2, 3\}$ we optimize the following objective,

$$\arg\min_{\theta_i} \sum_{(\mathbf{I}, \mathbf{S}) \in \mathcal{D}} \mathcal{L}\big(\mathrm{CF}^i_{\theta_i}(\mathbf{I}, \mathcal{T}_i), S\big). \qquad (3)$$

As training loss $\mathcal{L}(\cdot, \cdot)$, we minimize the mesh edge loss and Chamfer distance computed on point clouds of 150k points sampled from the predicted and ground-truth surfaces using random uniform sampling. The implementation of these losses and sampling algorithm are provided in the `PyTorch3D` library [56].

### 3.1 DMD Diffeomorphic Mesh Deformation module

The introduction of a Diffeomorphic Mesh Deformation module (DMD) is driven by the following classification of surfaces in 3 dimensions:

**Theorem 3.1.** *Suppose that $B$ is a smooth closed manifold of dimension $2$ embedded in $\mathbb{R}^3$. Suppose that $\Phi : [0, \tau] \times \mathbb{R}^3 \to \mathbb{R}^3$ is a family of homeomorphisms (continuous map such that for each $t$, the mapping $x \mapsto \Phi(t, x)$ is bijective with continuous inverse), with $\Phi(0; \mathbf{x}) = \mathbf{x}$. Then, for each $t$, the homotopy classes of $B$ and $\Phi_{\sharp B}(t) = \{\Phi(t, y), y \in B\}$ are the same.*

This theorem means that if $B$ is a sphere, the surface $\Phi_{\sharp B}(t)$ is of genus 0 with no self-intersection.

**Existence and uniqueness of a solution to the continuous problem**

The DMD generation of diffeomorphic mappping relies on the resolution of a continuous flow ODE. For that purpose, let $v : \mathbf{x} \in \Omega \mapsto v(\mathbf{x}) \in \mathbb{R}^3$ be a constant over time vector field supported on the MRI space $\Omega$ and obtained by tri-linear interpolation of a feature map $\mathbf{U} \in \mathbb{R}^{H \times W \times D \times 3}$. Suppose that $v = 0$ on $\partial\Omega$. The image origin is denoted by $O = (o_1, o_2, o_3)$ and denote by $d_i$ the interpolation spacing in the $i$-th direction such that $\Omega = [o_1, o_1 + d_1(H-1)] \times [o_2, o_2 + d_2(W-1)] \times [o_3, o_3 + d_3(D-1)]$.

**Theorem 3.2.** *Existence and uniqueness of the solution. Define $\Phi$ through the autonomous ODE,*

$$\frac{d\Phi(s; \mathbf{x})}{ds} = v\left(\Phi(s; \mathbf{x})\right), \; \text{with } \Phi(0; \mathbf{x}) = \mathbf{x}. \tag{4}$$

*Then $\Phi$ is uniquely defined on $\mathbb{R} \times \Omega$, is Lipschitz and for each $t$, the mapping $x \mapsto \Phi(t, x)$ is bijective with Lipschitz inverse. The proof of this result can be found in the supplementary material.*

Being Lipschitz is more difficult to achieve than being merely continuous. Less formally, Theorem 3.2 ensures that if $B$ is smooth, then for all $t$, the surface $\Phi_{\sharp B}(t)$ may, in the worst case, have kinks. If $\Phi$ is only continuous and not Lipschitz, then the surface $\Phi_{\sharp B}(t)$ might have cusps that are more irregular than kinks. Note as well that the choice of the interpolation technique used to generate $v$ is pivotal since it drives the regularity of the right-hand side of Equation (4). Indeed, $v$ Lipschitz's constant boundedness allows the definition of a solution to the continuous problem. More importantly, and as described in the next section, it rules the step-size to use for obtaining a stable numeric method.

**Numerical resolution of ODE**

The DMD module solves for each vertices' position a flow ODE defined in Equation (4), defined by $\Psi$ the numeric approximation of $\Phi$ by an explicit forward method, the invertibility of this discretisation is given by the following theorem

**Theorem 3.3.** *Let $v$ be $L$-Lipschitz. Define $\Psi$ as the Forward Euler approximation,*

$$\Psi(h, x) = x + hv(x). \tag{5}$$

$\Psi$ *is a Lipschitz homeomorphism for each $h < L^{-1}$.*

As a result, in combination with Theorem 3.2, the surface $\Psi_{\sharp B}(h)$ is smooth as long as $hL < 1$. We note that the stability condition $hL < 1$ is commonly used in Computational Fluid Dynamics [39]. Less formally, this estimate tells us that the less regular (high gradient) $v$ is, the smaller the integration step should be chosen.

*Proof.* Let $h < L^{-1}$ and denote $f : x \mapsto x + hv(x)$. Then $f$ is a Lipschitz mapping and it is sufficient to prove the bijectivity of $f$. The injectivity stems from, for all $x \neq y$,

$$\|f(x) - f(y)\| \geq \|x - y\| - h\|v(x) - v(y)\| \geq \|x - y\|(1 - hL) > 0.$$

The surjectivity comes from the fact that for each $y$, the mapping $g : x \mapsto y - hv(x)$ is $hL$-Lipschitz with $hL < 1$, hence is a contraction. By a fixed point theorem, $g$ admits a unique $x^\star$ solution to $g(x^\star) = x^\star$, or equivalently $y = f(x^\star)$. Hence, $f$ is surjective. $\qquad\square$

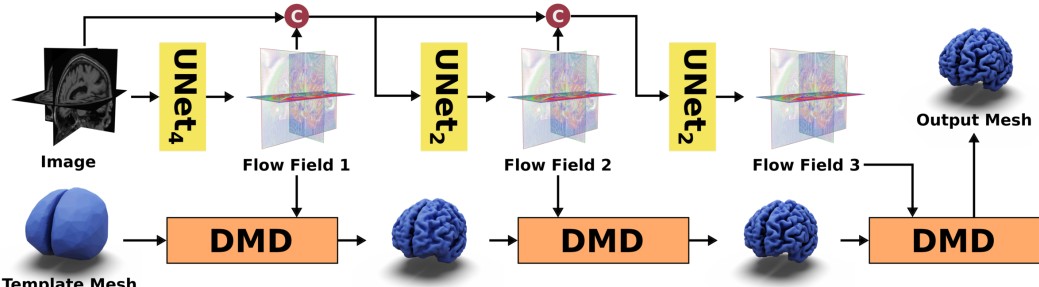

Figure 2: CorticalFlow architecture for the cortical surface reconstruction problem. Following the notations of Equation (2), $\text{UNet}^1_{\theta_1}$ implements a **UNet$_4$** layer and $\text{UNet}^{2,3}_{\theta_{2,3}}$ implements a **UNet$_2$** layer. The architecture of **UNet$_4$** and **UNet$_2$** are described in the supplementary materials.

---

**Algorithm 1** DMD module pseudo-code

---

**Input:** $\mathbf{U} \in \mathbb{R}^{H \times W \times D \times 3}$                $\triangleright$ Discrete flow
**Input:** $\mathcal{T}$ with vertices $(V_i)_{i \in 1..m}$         $\triangleright$ Triangle mesh
**Input:** $n \in \mathbb{N}^*$                 $\triangleright$ Number of integration steps
**Output:** Updated $(V_i)_{i \in 1..m}$
$h \leftarrow \frac{1}{n}$
**Ensure:** $h < \frac{1}{L}$
   **for** $i \in [\![1,m]\!]$ **do**
      **for** $j \in [\![1,n]\!]$ **do**
         $V_i \leftarrow \Psi(h, V_i)$
      **end for**
   **end for**

---

Theorem 3.3 ensures that $\Psi_{\sharp B}(h)$ is a non-intersecting manifold. Unfortunately, there exists another layer of numerical approximation that prevents $\Psi_{\sharp B}(h)$ to have the desired topological properties. Indeed, suppose that $B$ is a sphere, one never computes $\Psi_{\sharp B}(h)$, but triangulates a sphere with vertices $V_i$, and evaluates the image of the vertices using Algorithm 1. A new mesh is formed by using the image of the vertices and the original connectivity. Since the perturbed edges of the mesh are not the images of the original edges by the Forward Euler scheme, the resulting mesh may self-intersect (we refer interested readers to Section 2.3 of our supplementary material).

Notwithstanding this limitation, we use the rule of thumb $hL \leq 1$, and check that for all considered examples, and we have $hL \leq 1$.

## 3.2 Network architecture

As shown in Figure 2, CorticalFlow consists of a chain of three deformations. Note that more deformation modules could be used, but we focus on three modules to have a fair comparison with existing techniques. The first deformation module receives as input a volumetric image and outputs a flow vector field with the same dimensions using UNet-3D [59]. This discrete flow vector field is integrated by the DMD module to compute smooth deformations as explained in Section 3.1. The subsequent UNet-3D receives as input the image and the flow vector fields predicted by the previous deformation modules. The set of resulting mappings are composed to produce the final mesh.

For training CorticalFlow, we adopted a sequential approach where we train one deformation module at a time while freezing the weights of the others. We train the first deformation with a low-resolution template ($\mathcal{T}_1$) of 30k vertices and a UNet-3D architecture with four down/up-sampling levels. To train the second and third deformations, we increase the resolution of the template mesh to 135k and 435k vertices, for $\mathcal{T}_2$ and $\mathcal{T}_3$ respectively, and reduce the UNet-3D down/up-sampling levels to two. These networks are respectively labeled as **UNet$_4$** and **UNet$_2$** in Figure 2 while their layer details are described in our supplementary material. The choice of the architecture depth is motivated by the numerical conditions on the integration step-size derived in Section 3.1.

To verify the conditions of Theorem 3.3, it is essential to mention the use of template meshes with different resolutions and different UNet architectures. Indeed, the first block has to provide a large deformation resulting in a high $\|v\|$. To keep the Lipschitz constant $L$ small, one observes that the use of a low-resolution template with 30k vertices along with a deeper convolutional architecture forces the UNet to recover only coarse details and thus produces a flow vector field with a small $\nabla v$. During the second and third deformation, more details and higher resolution folds can be learned,

with templates composed of $135\text{k}$ and $435\text{k}$ vertices, respectively (see the ablation study presented in Section 1.1 of our supplementary material). We empirically verify that this hierarchy of deformations was beneficial for producing a flow vector field $v$ with a small Lipschitz constant. This multi-step approach allows attaining up to $14.5$ times less self-intersection in comparison with Neural Mesh Flow (see Table 1).

To generate the template mesh, we take the convex hull of all surfaces contained in the training dataset and remesh them uniformly using JIGSAW [21]. To achieve a different order of refinement, we use the midpoint subdivision algorithm implemented in MeshLab [12]. Model hyper-parameters and further implementation details are provided in the supplementary material.

## 4 Experiments

We benchmark CorticalFlow and other existing deep learning techniques on the cortical surface reconstruction problem. The goal is to estimate geometrically accurate and topologically correct triangular meshes for the inner and outer cortical surfaces from a given MRI. Like previous works [14, 16, 31, 34, 38, 61], these surfaces are further divided into the left and right brain hemispheres. See below a summary of the dataset, evaluated methods, and metrics used in our benchmark, in addition to the detailed discussion of the results summarized in Table 1.

| | Left Outer Surface | | | | | | Right Outer Surface | | | | | |
|---|---|---|---|---|---|---|---|---|---|---|---|---|
| | CH(mm)↓ | HD(mm)↓ | CHN↑ | %SIF↓ | DSC↑ | VS↑ | CH(mm)↓ | HD(mm)↓ | CHN↑ | %SIF↓ | DSC↑ | VS↑ |
| CorticalFlow | 0.681 | 0.802 | 0.932 | 0.686 | 0.977 | 0.993 | 0.693 | 0.815 | 0.929 | 1.239 | 0.976 | 0.994 |
| 1.148 sec / 3.071 GB | (±0.098) | (±0.049) | (±0.006) | (±0.469) | (±0.003) | (±0.006) | (±0.091) | (±0.046) | (±0.006) | (±0.629) | (±0.003) | (±0.005) |
| DeepCSR | 0.925 | 0.898 | 0.933 | 0.000 | 0.981 | 0.993 | 0.912 | 0.895 | 0.933 | 0.000 | 0.981 | 0.993 |
| 577.492 sec / 11.099 GB | (±0.265) | (±0.135) | (±0.011) | (±0.000) | (±0.003) | (±0.005) | (±0.208) | (±0.125) | (±0.010) | (±0.000) | (±0.003) | (±0.005) |
| NMF | 1.557 | 1.485 | 0.885 | 0.818 | 0.953 | 0.987 | 1.772 | 1.588 | 0.871 | 1.340 | 0.946 | 0.983 |
| 42.808 sec / 14.431 GB | (±0.200) | (±0.116) | (±0.008) | (±0.245) | (±0.003) | (±0.008) | (±0.177) | (±0.101) | (±0.005) | (±0.292) | (±0.005) | (±0.009) |
| QuickNAT | 3.732 | 2.582 | 0.874 | 0.000 | 0.956 | 0.975 | 4.282 | 2.800 | 0.870 | 0.000 | 0.954 | 0.971 |
| 13.003 sec / 9.627 GB | (±1.971) | (±1.011) | (±0.024) | (±0.000) | (±0.006) | (±0.011) | (±2.293) | (±1.085) | (±0.024) | (±0.000) | (±0.006) | (±0.011) |
| Voxel2Mesh | 7.188 | 3.893 | 0.721 | 0.857 | 0.896 | 0.976 | 7.461 | 3.955 | 0.717 | 0.833 | 0.888 | 0.967 |
| 20.840 sec / 23.400 GB | (±0.669) | (±0.218) | (±0.016) | (±0.260) | (±0.008) | (±0.011) | (±0.711) | (±0.227) | (±0.017) | (±0.260) | (±0.008) | (±0.010) |
| | Left Inner Surface | | | | | | Right Inner Surface | | | | | |
| | CH(mm)↓ | HD(mm)↓ | CHN↑ | %SIF↓ | DSC↑ | VS↑ | CH(mm)↓ | HD(mm)↓ | CHN↑ | %SIF↓ | DSC↑ | VS↑ |
| CorticalFlow | 0.608 | 0.785 | 0.941 | 0.033 | 0.962 | 0.987 | 0.599 | 0.783 | 0.942 | 0.030 | 0.962 | 0.987 |
| 1.148 sec / 3.071 GB | (±0.098) | (±0.060) | (±0.007) | (±0.030) | (±0.005) | (±0.010) | (±0.093) | (±0.059) | (±0.007) | (±0.029) | (±0.005) | (±0.010) |
| DeepCSR | 0.653 | 0.767 | 0.956 | 0.000 | 0.963 | 0.985 | 0.634 | 0.760 | 0.956 | 0.000 | 0.964 | 0.986 |
| 577.492 sec / 11.099 GB | (±0.138) | (±0.063) | (±0.006) | (±0.000) | (±0.006) | (±0.012) | (±0.139) | (±0.057) | (±0.006) | (±0.000) | (±0.006) | (±0.011) |
| NMF | 1.404 | 1.447 | 0.884 | 0.355 | 0.928 | 0.984 | 1.434 | 1.463 | 0.883 | 0.436 | 0.927 | 0.984 |
| 42.808 sec / 14.431 GB | (±0.185) | (±0.140) | (±0.011) | (±0.168) | (±0.006) | (±0.012) | (±0.184) | (±0.135) | (±0.010) | (±0.188) | (±0.006) | (±0.012) |
| QuickNAT | 2.370 | 1.836 | 0.906 | 0.000 | 0.905 | 0.924 | 2.295 | 1.788 | 0.908 | 0.000 | 0.906 | 0.931 |
| 13.003 sec / 9.627 GB | (±1.893) | (±0.734) | (±0.023) | (±0.000) | (±0.027) | (±0.031) | (±1.963) | (±0.725) | (±0.023) | (±0.000) | (±0.027) | (±0.031) |
| Voxel2Mesh | 5.968 | 3.461 | 0.703 | 0.621 | 0.850 | 0.982 | 6.172 | 3.549 | 0.707 | 0.761 | 0.842 | 0.971 |
| 20.840 sec / 23.400 GB | (±0.657) | (±0.247) | (±0.017) | (±0.220) | (±0.009) | (±0.014) | (±0.719) | (±0.272) | (±0.017) | (±0.271) | (±0.010) | (±0.018) |

Table 1: Cortical Surface Reconstruction Benchmark. Consider the evaluation metrics: chamfer distance (CH), Hausdorff distance (HD), chamfer normals (CHN), percentage of self-intersecting faces (%SIF), Dice Score (DSC), and Volume Similarity (VS). ↓ indicates smaller metric value is better, while ↑ indicates greater metric value is better. We also report the inference runtime and GPU memory footprint required by the compared algorithms.

**Dataset.** We used the same MRIs, pseudo-ground-truth surfaces, and data splits as [61]. This dataset consists of 3,876 MRI images extracted from the Alzheimer's Disease Neuroimaging Initiative (ADNI) [36] and their respective pseudo-ground-truth surfaces generated with the FreeSurfer V6.0 cross-sectional pipeline [25]. We train all methods on the training set ($\approx 60\%$) until their losses plateau on the validation set ($\approx 10\%$) and report their performance on the test set ($\approx 30\%$). We refer the reader to [61] and our supplementary material for full details on the dataset.

**Evaluated Methods.** We compare CorticalFlow to the following methods: DeepCSR[1] [61], Voxel2Mesh[2] [73], NMF[3] [30], and QuickNAT[4] [60]. As discussed in Section 2, DeepCSR is the state-of-the-art geometric deep learning model for cortical surface reconstruction, while Voxel2Mesh is a deformation-based model proposed to retrieve generic anatomical surfaces from volumetric medical images like MRIs and CT scans. Differently, NMF is a deformation-based model for single-view object reconstruction from 2D images. To adapt this model to our task whose input is a

---

[1]DeepCSR official implementation retrieved from `https://bitbucket.csiro.au/projects/CRCPMAX/repos/deepcsr`

[2]Voxel2Mesh official implementation retrieved from `https://github.com/cvlab-epfl/voxel2mesh`

[3]NMF official implementation retrieved from `https://github.com/KunalMGupta/NeuralMeshFlow`

[4]QuickNAT official implementation retrieved from `https://github.com/ai-med/quickNAT_pytorch`

3D MRI, we evaluate different 3D convolutional network backbones based on UNet [59], ResNet [33], and Hypercolumn [61]. The Hypercolumn backbone provides the best results thanks to its vertex-dependent features. As such, this is used as the NMF backbone in our benchmark. At the same time, the results for the other backbones are presented in our supplementary material. We also evaluate a baseline composed of a state-of-the-art brain segmentation model QuickNAT [60] followed with the marching cubes to evaluate the surface. All of these methods were trained and evaluated using a NVIDIA P100 GPU and Intel Xeon (E5-2690) CPU, except Voxel2Mesh which required a NVIDIA RTX 3090 GPU due to its GPU memory requirements.

**Evaluation Metrics.** We compare these methods for their geometric accuracy and surface regularity, as well as their time and space complexity. As a measure of geometric accuracy, we report the standard Chamfer distance (CH), Hausdorff distance (HD), and Chamfer normals (CHN). We compute these distances for point clouds of 200k points uniformly sampled from the predicted and target surfaces. As a measure of regularity, we compute the percentage of self-intersecting faces (%SIF) using `PyMeshLab` [48]. We also report volumetric overlap metrics [67] including Dice Score (DSC) and Volume Similarity (VS) computed on the high-resolution rasterization ($4\times$the input MRI resolution) of the generated and ground-truth surfaces.

For the time and space complexity of the evaluated methods, we report their average inference time (in seconds) and inference GPU memory footprint (in GB) to reconstruct the four cortical surfaces, respectively.

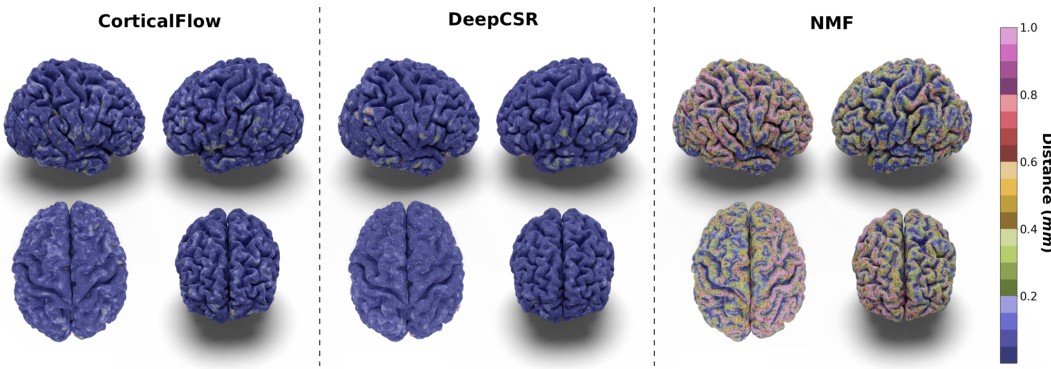

Figure 3: Predicted outer cortical surfaces color-coded with the distance to the pseudo-ground-truth surfaces. Here we present the results for the three best-performing methods in terms of geometric accuracy: CorticalFlow, DeepCSR, and NMF. See our supplementary materials for more examples.

**Results & Discussion.** In our experiments, we noticed that CorticalFlow produces more geometrically accurate surfaces than the other methods. On average, it presents better geometric metrics across all the cortical surfaces. In addition, as shown in Figure 3, CorticalFlow errors are smaller ($\leq 0.2mm$) and evenly spread across the surface compared to the other methods. In contrast, NMF and DeepCSR can present substantial errors ($\geq 1mm$). The former has its error spread across the entire surface, while the latter can produce large errors at specific regions.

CorticalFlow is also more robust than the competitors presenting lower error variation across individuals as suggested by the smaller standard deviation of the geometric metrics computed. Interestingly, CorticalFlow is also more robust to MRI artifacts even when the pseudo-ground-truth surface has poor quality. For instance, in Figure 4, CorticalFlow predictions are still plausible for a blurry input MRI while FreeSurfer fails significantly to generate appropriate surfaces for the same input. These examples support our claim that a regular parametrization allows us to reduce non-plausible and non-diffeomorphic predictions that our model cannot learn by construction.

CorticalFlow also generates triangular meshes with better properties than the evaluated methods. Compared to the deformation-based methods NMF and Voxel2Mesh, CorticalFlow predicted meshes are genus-zero surfaces and present a lower percentage of self-intersecting faces (mainly for the inner cortical surfaces). Figure 5a presents examples of self-intersecting faces produced by CorticalFlow, which are contrasted with the NMF predicted mesh for the same input MRI. The implicit-surface-

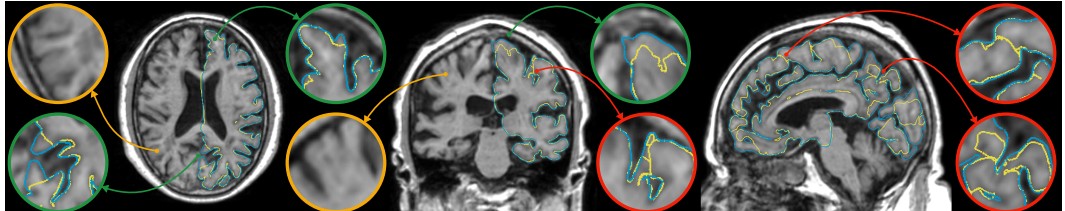

Figure 4: Slices of a blurry MRI and the outer surface delineation generated by FreeSurfer V6 (yellow contour) and CorticalFlow (blue contour). Orange circles highlight blurry MRI regions, green circles highlight FreeSurfer's underestimated areas, while red circles highlight non-plausible predictions avoided by CorticalFlow thanks to the diffeomorphism of its predicted deformations.

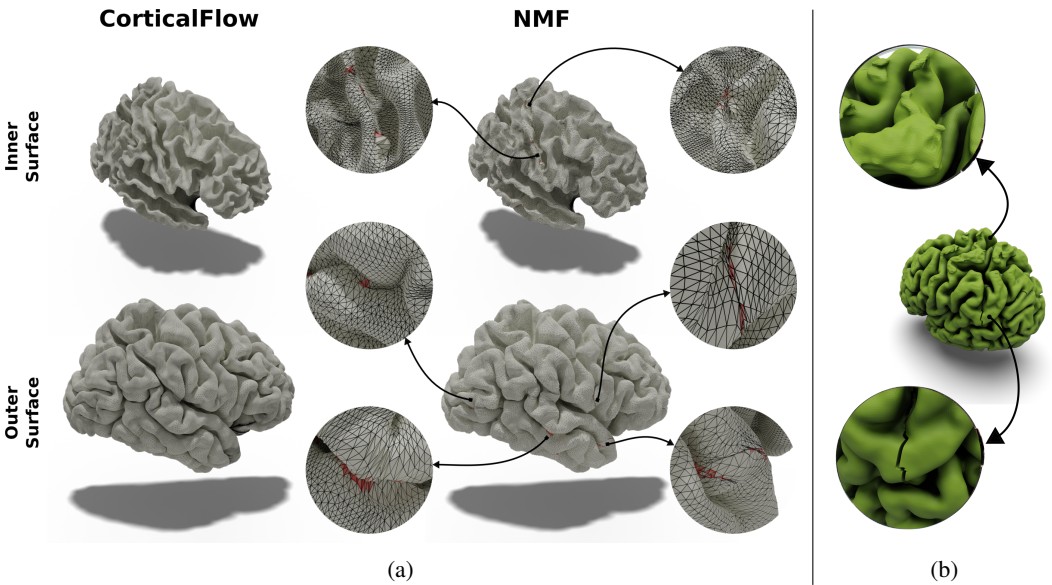

Figure 5: a: Predicted cortical surfaces by CorticalFlow and NMF with self-intersecting faces highlighted in red. b: Significant mistakes generated by the topology correction algorithm used in the DeepCSR method.

based DeepCSR method does not produce a single self-intersecting face since it employs computationally expensive post-processing routines like topology correction [4] and iso-surface extraction. However, these post-processing routines do not take into account the input MRI which can generate non-plausible corrections on the output mesh as previously observed in Segonne et al. [63] and exemplified in Figure 5b. Similarly, the voxel-wise segmentation baseline (i.e., QuickNAT) is free of self-intersecting faces, but it does not produce genus-zero surfaces. Indeed, QuickNAT's predicted surfaces are composed of multiple connected components presenting many handles and holes which is not acceptable for the purpose of cortical surface reconstruction. Some examples of QuickNAT reconstructed cortical surfaces are presented in our supplementary material. Therefore, we argue that CorticalFlow is the method of choice to reconstruct regular surfaces from volumetric images.

Due to its elemental construction (three UNet-3D backbones and an interpolation module for the integration), CorticalFlow remains highly efficient. It has a minimal GPU memory footprint and faster inference runtime while handling larger surfaces with more vertices both during training and inference. This feature allows its deployment on low-end computers and embedded devices which is pivotal in many scenarios across public health and for commercialization of affordable AI healthcare solutions [13, 55].

Finally, as a by-product of CorticalFlow's deformable and diffeomorphic nature, one can seamlessly obtain a sub-voxel resolution segmentation by applying a voxelization engine. This can capture variations below the image resolution while traditional segmentation methods [60] are restrained from working at the image resolution (see Figure 6a). Additionally, an essential component of

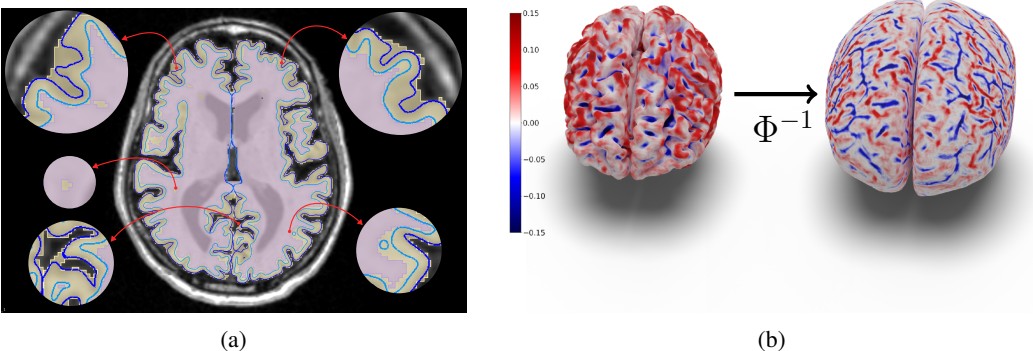

(a)                                          (b)

Figure 6: a: QuickNAT segmentation (inner surface volume in pink and outer surface volume in yellow) and CorticalFlow generated surfaces (inner surface in light blue and outer surface in dark blue). Note that the highlighted artifacts in QuickNAT segmentation can be easily surpassed by CorticalFlow thanks to its continuous representation of the region of interest. b: Outer cortical surface color-coded with the curvature of its cortical foldings. Our method allows us to map back this measure and any other biomarker to the input template using the composition of the inverse deformations $\Phi^{-1}$.

computational neuro-anatomy consists of computing local shape descriptors for different individuals and transferring them to the same reference space using conformal mappings [29, 66]. For the proposed model, one can efficiently compute the inverse transformation $\Phi^{-1}$ as shown in Figure 6b for the surface curvature descriptor.

## 5 Conclusion

This paper introduces CorticalFlow - a geometric deep learning model for efficiently reconstructing high-resolution, accurate, and regular triangular meshes from volumetric images. We develop a lightweight neural network to predict a dense 3D flow vector field from a volumetric image. Then, we describe a new Diffeomorphic Mesh Deformation (DMD) module, which is parameterized by a set of diffeomorphic mappings. This includes the derivation of numerical conditions for recasting the continuous flow ODE problem into an efficient discrete solver. Finally, we extensively verify that the proposed model achieves state-of-the-art performance in the challenging brain cortical surface reconstruction problem. This benchmark reveals that CorticalFlow is more accurate and, by construction, more robust to image artifacts providing anatomically plausible surfaces. Thanks also to its low space and time complexity, the proposed method can facilitate large-scale medical studies and support new healthcare applications.

## 6 Compliance with Ethical Standards

This research was approved by CSIRO ethics 2020 068 LR.

## 7 Acknowledgements

This work was funded in part through an Australian Department of Industry, Energy and Resources CRC-P project between CSIRO, Maxwell Plus and I-Med Radiology Network.

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
