# OpenReview forum: "CorticalFlow: A Diffeomorphic Mesh Transformer Network for Cortical Surface Reconstruction"
_NeurIPS.cc/2021/Conference — NeurIPS 2021 Poster_

### Official Review · Reviewer_wgwm · 2021-07-16

**Rating:** 7
**Confidence:** 4

**Summary:**

The paper proposes a template registration approach for meshes that is fast to compute, and that maintains constraints on the topology of the mesh / structure to be segmented. Applications on cortex segmentation in 3D images are demonstrated.

**Limitations And Societal Impact:**

Limitations: are addressed

Societal impact: does not apply

**Main Review:**

Plus:
* well embedded into the existing literature
* well chosen and fair baselines
* still: good and competitive results, qualitatively and quantitatively

Con:
* theorem 3.3 assumes v is L-Lifshitz, while v is only a trilinear interpolation (L172) and, hence, not differentiable at the voxel boundaries. This would imply L is infinite... which may be seen as a undesired property. (Could you choose h so that h L <1?) Please comment.
* a quantitative comparison with "pseudo-ground truth" is dangerous - you may only reproduce the FreeSurfer bias better than others do.
* I might have missed it but can you comment on the loss function? (Two sided chamfer distance?)
* shapeflow would be worth to be mentioned. I doesn't start with images, so it doesn't have to be added to the baseline comparison, but its concept is similar and should be discussed.
* to me theorem 3.1 is not obvisous. Would it be worth to present a proof?

**Time Spent Reviewing:**

2.5

---

> ### Author Response · Authors · 2021-08-10
> **Answer to reviewer wgwm**
>
> * *theorem 3.3 assumes v is L-Lifshitz, while v is only a trilinear interpolation (L172) and, hence, not differentiable at the voxel boundaries. This would imply L is infinite... which may be seen as a undesired property*
>
> A function $f :  \mathbb{R}^3 \rightarrow \mathbb{R}^3$ is said to be $L$-Lipschitz if
>
> $ \forall (x,y) , \quad  \lVert f(x) - f(y) \rVert \leq L \lVert x - y \rVert. $
>
> We agree with the reviewer that a differentiable function on a bounded domain is Lipschitz if and only if its derivatives are bounded, but this does not mean that a non-differentiable function cannot be Lipschitz. For instance, the absolute value (not differentiable at 0) is 1-Lipschitz.
>
> * *A quantitative comparison with "pseudo-ground truth" is dangerous - you may only reproduce the FreeSurfer bias better than others do.*
>
> We agree that further experiments would be necessary to assess the usefulness of the generated surfaces when used for brain morphometry and statistical analyses. This paper aimed to compare our method in the benchmark proposed in [54]. We conclude that CorticalFlow was as competitive as DeepCSR in terms of geometric accuracy but reduce its computational cost massively from 9 minutes to 1 second and tends to generate fewer artifactual patterns. The evaluation of morphometric biomarkers stemming from the produced surfaces is the subject of future work.
>
> * *I might have missed it but can you comment on the loss function? (Two sided chamfer distance?)*
>
> To learn the weights of our model, we minimise well-known surface mesh losses like the Chamfer distance and mesh edge loss between our predicted surface and its respective target surface. For both loss functions, we used their implementation provided by the third-party library PyTorch3D; this point is discussed in the supplementary material but will be moved in the main manuscript.
>
> * *Shapeflow would be worth to be mentioned. I doesn't start with images, so it doesn't have to be added to the baseline comparison, but its concept is similar and should be discussed.*
>
> We will include the reference to Shapeflow in the final version of the manuscript.
>
> * *to me theorem 3.1 is not obvisous. Would it be worth to present a proof?*
>
> This is a standard result; we will include a reference to *Differential Topology of M.W Hirsch*

---

### Official Review · Reviewer_xE8n · 2021-07-21

**Rating:** 7
**Confidence:** 4

**Summary:**

This paper focuses on the **segmentation of the cortex** in 3D brain MRI volumes. The proposed method combines a **diffeomorphic registration layer** with **volumetric U-Net** modules to iteratively register a (very smooth, sphere-like) template mesh onto the target “level set” of the 3D MRI. Performances are evaluated on the **ADNI dataset,** using FreeSurfer results as pseudo ground truth.


**Limitations And Societal Impact:**

The limitations and societal impact of this work have been adequatly addressed in the paper.

**Main Review:**

This paper is well-written, with clear attention paid to details. **Figures are well-chosen, informative and of the highest quality.** Performances on the ADNI dataset are highly competitive with a relatively simple architecture, fast run times and strong guarantees on the topology of the output surfaces. The generalisation capabilities of the trained model remain to be investigated, but there is no doubt that the present work is of **significant interest to the field of computational neuroanatomy.**


Nevertheless, I have two reservations about “warmly” recommending this paper for acceptance at NeurIPS 2021:

1 — The paper brings relatively **few novel ideas** to the field.

In Section 3.1, the authors present their “Diffeomorphic Mesh Transformer” layer as a fundamental contribution. (I would suggest replacing “Transformer” by e.g. “Deformation” to avoid confusions with “attention” modules.)

However, the content of this part of the paper has been **standard in the field of computational anatomy for about ten years.** The generation of diffeomorphisms for images and meshes through the integration of stationary (resp. time-varying) vector fields is usually known as the Stationary Velocity Fields “SVF” (resp. Large Deformations Diffeomorphic Metric Mapping “LDDMM”) framework.
I refer the authors to e.g. Chapter 5 of Jean Feydy’s PhD thesis (“Geometric data analysis, beyond convolutions”, 2020) for a recent overview of the topic and to e.g. the Deformetrica software for a user-friendly implementation of such methods on 3D meshes.

In the modern literature, “Diffeomorphic Mesh Transformers” are usually referred to as standard **“SVF exponentials” and used without problems on both images and meshes.** In this context, the main originality of the present paper lies in its network architecture, which is novel but **in line with recent works in the field** — see e.g. “Networks for Joint Affine and Non-parametric Image Registration“, Shen et al, (CVPR 2019) for a similar combination of modules in the case of image registration.

I appreciate the fact that this paper makes a **strong case for a deep diffeomorphic approach** to cortex segmentation, but **I would not overstate the significance of this work:** it does not bring much on the theoretical side, and does not provide an “outstanding” experimental evaluation on e.g. multiple datasets that would unambiguously establish the proposed algorithm as a reference method in the field of neuroanatomy.

2 — I do not know if this paper is a **right fit for NeurIPS.**

The present work is **clearly solid and useful to practitioners:** I have no doubt that this paper would be very well received at e.g. MICCAI (possibly as an oral), IPMI or in a neuroimaging journal. On the other hand, this work does not really bring forward any deep methodological idea for the wider community in shape analysis and computer vision.

All in all, this work reads like an **excellent application paper:** I personally enjoyed reading it and believe that “grounded” applicative works **bring a lot of value** to generalist conferences such as NeurIPS. On the other hand, I understand that the organizing committee may also want to keep the conference focused on papers that have a broader appeal to the wider ML community.


**Time Spent Reviewing:**

4

---

> ### Author Response · Authors · 2021-08-10
> **Answer to reviewer xE8n**
>
> * *In Section 3.1, the authors present their “Diffeomorphic Mesh Transformer” layer as a fundamental contribution. (I would suggest replacing “Transformer” by e.g. “Deformation” to avoid confusions with “attention” modules.)*
>
> We will change  “transformer” for deformation; as wisely noted by the reviewer, this naming can be confusing.
>
> * *In the modern literature, “Diffeomorphic Mesh Transformers” are usually referred to as standard “SVF exponentials” and used without problems on both images and meshes. In this context, the main originality of the present paper lies in its network architecture, which is novel but in line with recent works in the field — see e.g. “Networks for Joint Affine and Non-parametric Image Registration“, Shen et al, (CVPR 2019) for a similar combination of modules in the case of image registration.*
>
> We agree that the SVF framework has been massively used by the biomedical-DL community for image registration and using the scaling and squaring algorithm [1]. Concerning its use for meshes alignment, we only found the reference [13] that has the shortcomings described in our answer to the reviewer EX4h, which motivated us to devise an algorithm for points in arbitrary position (not on a regular grid).
>
> Concerning the nature of the task addressed, we wish to remind the reviewer that the devised method applies to anatomical surface reconstruction where Voxel2Mesh, DeepCSR, and CorticalFlow receive only a single MRI as input. This approach is slightly different from the image registration task, where two anatomies are put in correspondence (and where we suppose such a diffeomorphic mapping exists). In the field of neuroanatomy, a non-registration based approach could have more significant outreach in the study of diseases where registration turns to be difficult as for the study of early brain development or for patients with strokes where significant discrepancies between the cortical folding do not allow the creation of a reliable atlas. The rough template surface that utilises CorticalFlow does not have such strong assumptions on the underlying anatomies.
>
> Thanks for bringing to our attention the work of Shen et al, which will be added to the final version of our manuscript.
>
> * *not provide an “outstanding” experimental evaluation on e.g. multiple datasets that would unambiguously establish the proposed algorithm as a reference method in the field of neuroanatomy.*
>
> This work motivates the use of a diffeomorphic mesh deformation module and compares its performances on a pre-established benchmark proposed in [54]; we emphasise the byproducts provided by a diffeomorphic SVF framework and try to showcase a few examples where this method could be useful. We agree with the reviewer that more work needs to be done to validate the clinical usefulness of such methods as transferability to other datasets and other diseases, data efficiency of the proposed algorithm (how many images are required to build and train a model) and the statistical analysis of morphometric biomarkers stemming from the produced surfaces, which is the subject of future work.

---

> > ### Comment · Reviewer_xE8n · 2021-08-13
> > **Reply to the authors**
> >
> > I thank the authors for their answers to all of the reviewers’ comments. In light of the unanimously positive comments, **I have no doubt that this good paper will be accepted at NeurIPS 2021**.
> >
> > For the sake of clarity, let me just stress (again) that using an ODE integrator on **mesh** vertices to generate diffeomorphic transformations is far from being a novel idea in the field of computational anatomy. It has been applied for more than a decade to SVF integrators and the more general LDDMM framework, with influential results.
> >
> > The authors are visibly unaware of the fact that these numerical schemes are routinely discussed at e.g. the MFCA and ShapeMI workshops, which are organized every other year at MICCAI conferences - arguably a more suitable venue for this work than NeurIPS.
> >
> > For a historical perspective on the subject, the authors may read e.g. the PhD theses of:
> > - Joan Glaunes (2005, https://helios.mi.parisdescartes.fr/~glaunes/TheseGlaunes.pdf with e.g. https://helios.mi.parisdescartes.fr/~glaunes/preprints/glaunes_cvpr04.pdf for a paper in English).
> > - Stanley Durrleman (2011, https://tel.archives-ouvertes.fr/tel-00631382/PDF/Durrleman_thesis.pdf) - who later went on to develop the Deformetrica software.
> > - Nicolas Charon (2014, https://tel.archives-ouvertes.fr/tel-00942078/document) - with registrations of e.g. a smooth sphere to a Stanford Bunny.
> > - Jean Feydy (2020, https://tel.archives-ouvertes.fr/tel-02945979/document).
> >
> >
> > In light of these references, the authors could certainly **re-focus their statement of contributions on their (excellent) application work**. The fact that the proposed method is applied to a slightly different task in neuro-anatomy does not provide sufficient ground for the strong claim of novelty in Sections 2 and 3.

---

> > > ### Comment · Area_Chair_kKLw · 2021-08-17
> > > **Thx for opening the discussion**
> > >
> > > Overall, this paper obtained a very positive assessment in the first round, and there seems to be mostly minor disagreement about the state of the art: previous contributions should be better acknowledged. At least this stands out from the latest comment of reviewer xE8n.
> > > Are we converging toward acceptance ? This is at least my impression. Please voice your opinion if you think otherwise.

---

> > > ### Author Response · Authors · 2021-08-18
> > > **Answer to Reviewer xE8n**
> > >
> > > The authors wish to thank the reviewer for his answer and supportive comments.
> > >
> > > We are aware of this literature and we will refocus our statement in the final version of our paper. In our last comment we were referring to the deep-learning-based approaches and their pervasive use of the scaling and squaring method which is not suitable for meshes. We will refer with more details to the different approaches for generating diffeomorphisms in Section 2.2. and in the introduction of Section 3 of our manuscript. In our claims, more emphasis will be added towards the numerical efficiency and flexibility of our module, i.e., a reduced memory footprint that allows the composition of three deformations on high-resolution meshes (450k vertices). We will also discuss the potential broader impact of this method in other branches of deep learning.

---

### Official Review · Reviewer_4s5o · 2021-07-21

**Rating:** 7
**Confidence:** 2

**Summary:**

This paper introduces CorticalFlow – a geometric deep learning system to reconstruct topologically correct inner and outer cortical surface meshes from 3D MRI images. The model possesses two blocks: 1) a UNet architecture that given a 3D MRI image predicts a flow vector field and 2) a diffeomeorphic mesh transformer (DMT) that given the flow vector field, outputs a diffeomorphic mapping that may be applied to an unfeatured template mesh. The overall architecture has three stages at increasing mesh resolution, trained separately. The resulting surfaces lack geometric artifacts seen in other approaches. The inference time is also significantly faster than the baseline methods with lower memory footprint.

**Limitations And Societal Impact:**

The authors should add more discussion on the limitations of their method.

**Main Review:**

Overall, the results are very impressive (Table 1, Figures 3,4,5). The method even improves upon FreeSurfer, the software used to generate the pseudo-ground truth surfaces, in cases where the underlying MRI is blurry/has artifacts (Figure 5). I have a few minor comments/clarifications to the authors:

* If the only allowable transformations are diffeomorphic, why are there self-intersecting faces (nonzero SIF% in Table 1)?
* For clarity, the authors should bold the best values in Table 1.
* Are some of the arrows incorrect in Figure 5? (0 arrows from Cortical Flow, 2 arrows from NMF)
* How do the number of parameters in the model compare to the parameters in the voxel representation?
* The authors should discuss potential applications or connections to other areas for a general ML audience
* The authors should discuss limitations of this method


**Time Spent Reviewing:**

3

---

> ### Author Response · Authors · 2021-08-10
> **Answer to reviewer 4s5o**
>
> * *If the only allowable transformations are diffeomorphic, why are there self-intersecting faces (nonzero SIF% in Table 1)?*
>
> When applied on the vertices of the meshes, Diffeomorphic transformations do not preclude having self-intersecting faces. We detail this issue in the last part of our supplementary material. One point worth mentioning is that the more refined the triangle meshes are, the less likely faces can self-intersect.
>
> * *How do the number of parameters in the model compare to the parameters in the voxel representation?*
>
> The number of voxels in the considered application is 96×192×160, which corresponds to an MRI with a 1mm isotropic resolution centred on one hemisphere of the brain (see Lines 65-83 of our supplementary material). Cortical flow has in total 519 865 learnable parameters distributed across the three deformation modules employed and trained sequentially, as explained in Section 3.2 of our paper.
>
> * *The authors should discuss potential applications or connections to other areas for a general ML audience / The authors should discuss limitations of this method.*
>
> In our paper, the computation of the stationary vector field is performed explicitly, and to integrate $v$, resampling on the exact 3D location of the vertices is performed. It requires having a dataset where the 3D information is located coherently with the ground-truth triangle mesh. Therefore, tasks with missing depth information, such as single-view reconstruction, are unsuitable for such architecture. However, tasks such as surface reconstruction from point-cloud or surface-only registration could be in the range of potential applications for the DMT module.
> As limitations, we reiterate that the devised method is tailored to reconstruct surfaces with a single connected component without holes or handles, like cortical surfaces. For generic objects composed of multiple parts, other approaches are more suitable. In addition, our reconstruction’s quality depends on how the training surfaces accurately describe the underlying 3D images and the quality of the provided templates.

---

### Official Review · Reviewer_EX4h · 2021-07-22

**Rating:** 6
**Confidence:** 4

**Summary:**


The paper presents a diffeomorphic surface fitting framework. The method takes in a volume and fits a surface by predicting stacked diffeomorphic flow fields, interpolating these to warp a template mesh in a stacked fashion, and warping to obtain the surface required. The authors use a scale-space/pyramid of deformations to obtain accurate deformations, and perform a series of experiments showing that in some metrics the method outperforms existing strategies.

**Limitations And Societal Impact:**

The authors don't seem to discuss these. I provide several ideas for experiments to extract a clarification of what the contributions are -- these experiments will hopefully also indicate some of the limitations.

**Main Review:**


It is hard for me to discern the explicit contribution of the papers. The authors review literature and note several methods that are very closely related. As far as I can tell, there are three parts to the method: a volume-to-SVF-network (which of course has been widely done), a SVF-to-mesh Transformer (which I don't fully understand, more below), and a multi-scale approach to this (which of course has been widely done). Perhaps another contribution the authors claim are the various Theorems in 3.1?

My best estimate is that the authors are claiming 3.1 (the mesh transformer and related theorems) as the main contribution, and perhaps putting the framework together. Unfortunately, although I focused on 3.1 quite a bit, I am having a hard time understanding a couple of things. The authors intersperse Theorems with some practical implementations -- in the sense that the theorems are nice but cannot be executed in practice where approximations are needed, but what the authors do in practice is a bit of a mystery to me still.

I strongly encourage the authors to clarify this section, perhaps separating theory from practical implementation/execution. I assume that lines 185-186 and Eq (4) are what the authors do to approximate the ODE, but it's unfortunately unclear to me how the V_k^i are estimated for k > 0 -- is it simply with tri-linear interpolation on demand? If so, how is this linear approximation more appropriate than scaling and squaring, which has been done for surfaces with networks (see e.g. [13]) and of course for volumes by many methods including several of those cited). To me, this is a crux of the method -- is there a practical contribution in this section, or is the contribution in combining the 3 parts of the framework and the theorems in 3.1? While I appreciate the Theorems, they do not seem like a substantial contribution by themselves.

In the experiments, as far as I can tell the authors implement a thorough set of experiments, and I appreciate the separation of train/val/test and including of error/variance measures. However, the results of CorticalFlow are very close to DeepCSR in most measures, perhaps other than CH. Interestingly, they are quite close in terms of HD, which is in some sense close to CH in what it tried to capture. Since the improvements are minimal, the authors analyze the specific types of improvements a bit more qualitatively -- that the improvements are more in robustness (no/fewer outliers) and when zooming in on regions they find instances where they have better topological fitting. I think what would be interesting or important is to really evaluate where the method contributes to insight -- which part of the framework is important, is it the ODE approximation? If so, can they replace it by a standard method that is currently used (e.g. scaling and squaring)? Is it the pyramid/scale-space (I'm not sure if the other methods like DeepCSR do this). I believe investigating this is crucial to the proposed method and would make the paper much stronger.

Overall, I think there is potential in the method, but I have a hard time understanding the root of the contribution -- is there an interesting insightful nugget, or is the main contribution (other than 3.1 Theorems) that the authors smartly put the pieces together (which is fine, but different than if the DMT is somehow completely novel). This can be better explained in the intro (after explaining the literature by having a list of contributions), clarified in 3.1 (by clarifying the DMT model in practice, how it's different from other SVF integrations that are used, and separating the Theorems) and in the experiments (via ablation studies or proper analysis). It seems to me that it is central to the paper to take these steps to clarify to the reader what the take-away is and what we are learning from this.

**Time Spent Reviewing:**

3

---

> ### Author Response · Authors · 2021-08-10
> **Answer to reviewer EX4h**
>
> * *I strongly encourage the authors to clarify this section, perhaps separating theory from practical implementation/execution. I assume that lines 185-186 and Eq (4) are what the authors do to approximate the ODE, but it's unfortunately unclear to me how the V_k^i are estimated for k > 0 -- is it simply with tri-linear interpolation on demand?*
>
> This equation describes one step of the explicit-Euler method (forward-Euler, a first-order explicit scheme), and $v$ is the tri-linear interpolation of the output of a UNet. $V$ is a discretisation of the solution of (3) with a step size h for the template mesh vertices’.
>
> * *If so, how is this linear approximation more appropriate than scaling and squaring, which has been done for surfaces with networks (see e.g. [13]) and of course for volumes by many methods including several of those cited). To me, this is a crux of the method -- is there a practical contribution in this section, or is the contribution in combining the 3 parts of the framework and the theorems in 3.1? While I appreciate the Theorems, they do not seem like a substantial contribution by themselves.*
>
> Our numerical algorithm for predicting diffeomorphic transformations differs significantly from VoxelMorph. Voxelmorph obtains the surface vertices displacements by first integrating the predicted flow field on the entire image (regular grid aligned with voxels centres) using the standard scaling and squaring procedure and then linearly interpolates this mapping for the vertices’ positions. While this approach seems to work well on registration problems where only small displacements are required, interpolating a deformation grid neither yields a diffeomorphic mapping nor allows the computation of a tractable inverse. Differently, we obtain a diffeomorphic mapping for surfaces by directly integrating the predicted flow field for each vertices’ position. This approach is more suitable for large deformations and requires less computational resources, as discussed in Section 2.2. More specifically, its computation scales linearly with the number of vertices of the template mesh (at most 435k). In contrast, VoxelMorph's approach scales linearly with the number of voxels of the input image (almost 3M) and the number of vertices of the template mesh.
>
> * *However, the results of CorticalFlow are very close to DeepCSR in most measures, perhaps other than CH. Interestingly, they are quite close in terms of HD, which is in some sense close to CH in what it tried to capture. Since the improvements are minimal, the authors analyze the specific types of improvements a bit more qualitatively -- that the improvements are more in robustness (no/fewer outliers) and when zooming in on regions they find instances where they have a better topological fitting.*
>
> According to the benchmark metrics of DeepCSR [54], the 90th percentile is reported for the Hausdorff distance metric. If the Hausdorff distance is computed for the max (by randomly sampling 200k point clouds from the GT and the reconstructed surface), we have the following results:
>
> | method        |     Left Pial      |    Right Pial      |    Left White      |     Right White  |
> |------------------|:------------------:|:--------------------:|:--------------------:|:-------------------:|
> | DeepCSR    | 7.769 ± 1.227 |  7.787 ± 1.112  |  6.645 ± 1.367  |  6.463 ± 1.349 |
> | CorticalFlow| 5.932 ± 1.036 |  5.945 ± 1.019  |  5.273 ± 1.263  |  5.225 ± 1.351 |
>
> We display such pitfalls for DeepCSR in Figure 5-b.
>
> * *I think what would be interesting or important is to really evaluate where the method contributes to insight -- which part of the framework is important, is it the ODE approximation? If so, can they replace it by a standard method that is currently used (e.g. scaling and squaring)? Is it the pyramid/scale-space (I'm not sure if the other methods like DeepCSR do this). I believe investigating this is crucial to the proposed method and would make the paper much stronger.*
>
> As mentioned in our previous response, the task of cortical surface reconstruction and our choice of initial template surface is not similar to a registration framework and thence not suitable for scaling and squaring integration. An idea of multi-resolution is present in the DeepCSR model, where feature maps at different resolutions are combined in a hypercolumn vector. In addition, the effect of adding several UNET+DMT blocks is presented in the supplementary material through an ablation study.
>
> * *This can be better explained in the intro (after explaining the literature by having a list of contributions), clarified in 3.1 (by clarifying the DMT model in practice, how it's different from other SVF integrations that are used, and separating the Theorems) and in the experiments (via ablation studies or proper analysis). It seems to me that it is central to the paper to take these steps to clarify to the reader what the take-away is and what we are learning from this.*
>
> We will re-write the introduction according to the reviewer’s remarks. Concerning the integration, the stationary vector-field SVF framework to generate diffeomorphic transformation (solving the resulting IVP) is not new, as noted with our references to previous work [1,2,3,17]. At the crossroad with the Deep Learning field, numerous methods for image registration have been built upon this SVF framework and using scaling and squaring; in our paper, we cite a subset of these papers [13,41] and will include the reference provided by reviewer xE8n. This widely used integration method does not produce satisfactory deformations in the context of cortical surface reconstruction where triangle meshes’ vertices are densely packed in a limited number of voxels. In addition, the deformation range is larger than in a registration context. To alleviate these issues, we developed a novel integration technique that is particularly suitable when the diffeomorphisms have to be generated on a non-regular grid. We will clarify with more detail this contribution in the final version of our manuscript.

---

### Decision · Program_Chairs · 2021-09-28

**Decision:**

Accept (Poster)

**Comment:**

The paper presents a Geometric deep learning model that learns to diffeomorphic deform a regular
template mesh towards a targeted object.
The paper was found interesting from a methods perspective and it lead to impressive results. The
methodology is strong and elegant, making a strong case for a "deep diffeomorphic" approach. The
experimental baselines are well chosen.
The only controversy concerned the state of the art, that may not have been sufficiently well
acknowledged. Part of the contributions are grounded in a set of now classical contributions on LDDMM,
which was not made clear in the writing. The authors however received this comment well and should
address it.
In summary, there is a clear accept here. But the updated version should dedicate more space to
acknowledging prior work.

**Consistency Experiment:**

NeurIPS has a long history of experimentation. In 2014, NeurIPS ran an experiment in which 10% of submissions were reviewed by two independent committees to quantify the randomness in the review process. This year, we repeated a variant of this experiment to see how the quality of the review process has changed over time.  This paper was part of the experiment and was therefore assigned to two committees (consisting of reviewers, an Area Chair, and a Senior Area Chair) that reached independent decisions.  If both committees made the same recommendation, this recommendation was followed. If a single committee recommended acceptance, the paper was accepted (with the exception of a few cases in which the other committee identified what we considered a fatal flaw, e.g., an error in a key result).

This copy’s committee reached the following decision: **Accept (Spotlight)**

The other committee assigned to the paper recommended **Reject**.  You can find the other set of reviews, along with any follow up discussion with the authors here:
https://openreview.net/forum?id=wDI6CNTR3yP